# Functional Analysis of Steroidogenic Factor 1 (sf-1) and 17α-Hydroxylase/Lyase (cyp17α) Promoters in Yellow Catfish *Pelteobagrus fulvidraco*

**DOI:** 10.3390/ijms22010195

**Published:** 2020-12-27

**Authors:** Wu-Hong Lv, Guang-Hui Chen, Mei-Qin Zhuo, Yi-Huan Xu, Yi-Chuang Xu, Xiao-Ying Tan

**Affiliations:** Key Laboratory of Freshwater Animal Breeding, Ministry of Agriculture, Fishery College, Huazhong Agricultural University, Wuhan 430070, China; lvwuhong@webmail.hzau.edu.cn (W.-H.L.); cgh0626@webmail.hzau.edu.cn (G.-H.C.); zmq@mail.hzau.edu.cn (M.-Q.Z.); xuyihuan@webmail.hzau.edu.cn (Y.-H.X.); xuyichuang@webmail.hzau.edu.cn (Y.-C.X.)

**Keywords:** sf-1, cyp17α, steroidogenesis, transcriptional regulation, *Pelteobagrus fulvidraco*

## Abstract

The present study was performed to clone and characterize the structures and functions of steroidogenic factor 1 (sf-1) and 17α-hydroxylase/lyase (cyp17α) promoters in yellow catfish *Pelteobagrus fulvidraco*, a widely distributed freshwater teleost. We successfully obtained 1981 and 2034 bp sequences of sf-1 and cyp17α promoters, and predicted the putative binding sites of several transcription factors, such as Peroxisome proliferator-activated receptor alpha (PPARα), Peroxisome proliferator-activated receptor gamma (PPARγ) and Signal transducer and activator of transcription 3 (STAT3), on sf-1 and cyp17α promoter regions, respectively. Overexpression of PPARγ significantly increased the activities of sf-1 and cyp17α promoters, but overexpression of PPARα significantly decreased the promoter activities of sf-1 and cyp17α. Overexpression of STAT3 reduced the activity of the sf-1 promoter but increased the activity of the cyp17α promoter. The analysis of site-mutation and electrophoretic mobility shift assay suggested that the sf-1 promoter possessed the STAT3 binding site, but did not the PPARα or PPARγ binding sites. In contrast, only the PPARγ site, not PPARα or STAT3 sites, was functional with the cyp17α promoter. Leptin significantly increased sf-1 promoter activity, but the mutation of STAT3 and PPARγ sites decreased leptin-induced activation of sf-1 promoter. Our findings offered the novel insights into the transcriptional regulation of sf-1 and cyp17α and suggested leptin regulated sf-1 promoter activity through STAT3 site in yellow catfish.

## 1. Introduction

Steroid hormones are involved in the regulation of many physiological processes, such as embryonic development, sex differentiation, metabolism and reproduction in vertebrates [1], and their biosynthesis was regulated by key transcription factors and enzymes, such as steroidogenic factor 1 (sf-1) and 17α-hydroxylase (cyp17α) [2]. The sf-1 is an important transcription factor with critical regulatory roles in the transcription of steroidogenic genes [3,4]. cyp17α is the only enzyme responsible for the conversion of C21 steroids to C19 steroids in the pathway of steroidogenesis [1,2]. At present, studies have been conducted on the function of cyp17α and sf-1 on the mRNA and protein levels in several physiological processes [5,6]. In mammals, several transcriptional factors, such as nuclear transcription factor Y (NF-Y), specificity protein 1 (SP1) and cAMP-response element binding protein (CREB), positively regulate the activities of sf-1 promoter [7,8,9]. However, information associated with the structure and transcriptional regulation of sf-1 and cyp17α promoters were very scarce in fish.

Peroxisome proliferator-activated receptor alpha and gamma (PPARα and PPARγ) are the two important transcription factors that modulate the expression of many target genes involved in numerous physiological processes, including steroidogenesis [10,11,12]. Considering that cyp17α and sf-1 are also the key enzyme and transcription factor in regulating steroidogenesis, we hypothesize that PPARα and PPARγ regulated steroidogenesis by targeting cyp17α and sf-1.

Leptin belongs to a member of the cytokines and plays a vital role in reproduction [13]. Leptin has also important function in digestion (e.g., in birds), the connection with digestion was revealed by expression and immunohistochemistry analysis by Seroussi et al. [14]. Several studies reported the effects of leptin on the control of steroidogenesis in animals and cells [15,16], but the results were not consistent. For example, several studies reported that leptin attenuated gonadotropin or growth factor-stimulated steroidogenesis in isolated theca and granulosa cells [17,18], but other studies pointed out that leptin induced the synthesis of estrogen in ovary and ovarian follicles [19,20]. Signal transducer and activator of transcription 3 (STAT3) is identified as the most important factor to transduce the leptin signal [21,22,23]. In mammals, studies have suggested that leptin modulated steroidogenic by STAT3 [13,19]. Wu et al. [22] demonstrated that leptin regulated the transcriptional activities of lipid metabolism-related target genes by STAT3 in yellow catfish. However, whether leptin could directly mediate the transcriptional activities of genes involved in steroidogenesis, such as sf-1 and cyp17α, remained unknown.

Yellow catfish *Pelteobagrus fulvidraco*, an omnivorous freshwater teleost, has been cultured widely in China and other Asian countries due to the successful technology of artificial propagation. However, further investigation remained into the regulation of steroidogenesis during the propagation. It is well-known that the application of luciferase assay is a well-established technique in molecular biology for analysis of cloned promoter DNA fragments [24,25]. Thus, the present study was performed to characterize the regions of sf-1 and cyp17α promoters and investigated their transcriptional regulation in yellow catfish. Our study elucidated innovative insights into the regulatory mechanism for the biosynthesis of steroid hormones.

## 2. Results

### 2.1. Cloning and Sequence Analysis of sf-1 and cyp17α Promoters

In the present study, the transcription start sites (TSS) of sf-1 and cyp17α were identified, and 1981 bp and 2034 bp sequences were cloned for sf-1 and cyp17α promoters, respectively. Several core promoter elements closed to TSS were identified on the sf-1 promoter, such as one CCAAT box from −38 bp to −27 bp and three SP1 binding sites located at−130 bp to −120 bp, −118 bp to −108 bp and −16 bp to −7 bp, respectively (Appendix A). One PPARα binding site (at −413 bp/−396 bp), one PPARγ binding site (at −1396 bp/-1388 bp) and one STAT3 binding site (at −1106 bp/−1096 bp) were predicted on the region of sf-1 promoter, respectively (Appendix A).

On the cyp17α promoter, we predicted one TATA box located from −150 bp to −136bp, one GC box located at −55 bp to −42 bp and one SP1 binding site located at −53 bp to -43 bp, one PPARα binding site (at −881 bp/−864 bp), one PPARγ binding site (at −89 bp/−70 bp) and one STAT3 binding site (at −1669 bp/ −1660 bp), respectively (Appendix A).

### 2.2. The Relative Luciferase Activity of 5’-Deletion Assay of the Regions of sf-1 and cyp17α Promoters

Deletion of the sequences from −1981 bp to −1421 bp of sf-1 promoter significantly enhanced the relative luciferase activity. There is no significant difference after the deletion of the sequences from −1421 bp to −921 bp. The deletion of the sequences from −921 bp to −489 bp significantly reduced the relative luciferase activity of sf-1 promoter (Figure 1A). Deletion of the sequences from −2034 bp to −1402 bp of cyp17α promoter significantly decreased the relative luciferase activity. Subsequent deletion to −687 bp significantly increased cyp17α promoter activity, but the deletion of the sequences from −687 bp to −330 bp showed no significant difference on cyp17α promoter activity (Figure 1A).

### 2.3. Overexpression Analysis of Yellow Catfish PPARα, PPARγ, and STAT3 of the Regions of sf-1 and cyp17α Promoters in HEK293T Cells

To reveal the mechanism of PPARα, PPARγ, and STAT3 regulating sf-1 and cyp17α in yellow catfish, we transfected PPARα, PPARγ, and STAT3 plasmids into HEK293T cells for 24 h, and found that they were successfully over-expressed in HEK293T cells (Figure 2).

To study the activities of sf-1 and cyp17α promoters induced by PPARα, PPARγ, and STAT3, we co-transfected PPARα, PPARγ, and STAT3 plasmids along with the promoter constructs into HEK293T cells for 24 h and performed the 5’-deletion assay of sf-1 and cyp17α promoters, respectively. Overexpression of PPARα resulted in a marked reduction in sf-1 promoter activity by 39%, compared to the control. The sequence deletion between −1981 bp and +254 bp presented no significant influences on PPARα-induced promoter activity, indicating that negative response element to PPARα existed on −489/+254 bp region of sf-1 promoter (Figure 3A). Overexpressed PPARγ significantly increased the sf-1 promoter activity compared to the control, and showed no effect on the sequence deletion between −1981 bp and +254 bp on PPARγ-induced promoter activity, indicating that active response element to PPARγ might exist on −489/+254 bp region of sf-1 promoter (Figure 3B). Overexpression of STAT3 significantly decreased the promoter activity of sf-1, compared to the control, The sequence deletion between −1981 bp and −1421 bp completely abolished the inhibitory effect by STAT3, and sequence deletion between −1421 bp to −921 bp increased the promoter activity by STAT3. Nonetheless, the sequence deletion between −489 bp and +254 bp region of sf-1 promoter completely abolished the STAT3-induced upregulation, reflecting that the negative response elements existed at −1981/−1421 bp and −1421/−921 bp, and the positive response element existed at −921/−489 bp regions of sf-1 promoter to STAT3 (Figure 3C).

Overexpression of PPARα markedly reduced the promoter activity of cyp17α compared to the control. The inhibitory effect by PPARα was completely abolished when the sequence was deleted from −1402 to −687 bp. However, deleting the sequence between −687 bp and −330 bp recovered the inhibitory effect by PPARα, indicating that there are two negative response elements at −1402/−687 bp and −330/+68 bp, and one positive response element at −687/−330 bp region of cyp17α promoter to PPARα (Figure 4A). Compared to the control, overexpression of PPARγ increased the promoter activity of cyp17α. However, deleting the sequence between −2034 bp and −330 bp presented no significant effects on PPARγ-induced cyp17α promoter activity, indicated that there is one positive responsive element at −330/+68 bp region of cyp17α promoter to PPARγ (Figure 4B). Overexpression of STAT3 enhanced the cyp17α promoter activity by 3.1-fold compared to the control, and deleting the region between −2034 bp and −1402 bp of cyp17α promoter further up-regulated the stimulative effect of STAT3 overexpression, but the subsequent sequence deletion from −1402 bp to −687 bp alleviated this stimulative effect. These results indicated there are two positive responsive elements at −1402/−687 bp and −330/+68 bp region of cyp17α promoter to STAT3 (Figure 4C).

To investigate the response of promoters induced by leptin, we used 200 ng/mL leptin to incubate HEK293T for 24 h and performed the 5′ deletion assays. Leptin incubation significantly increased sf-1 promoter activity. In the leptin-treated groups, the sequence deletion between −1981 bp and −921 bp of sf-1 promoter showed no significant influences on luciferase activity. However, further deletion from −921 to −489 bp significantly down-regulated leptin-induced sf-1 promoter activity (Figure 5A). Leptin incubation show no effect on the cyp17α promoter activity, and no differences were found in relative luciferase activity of cyp17α promoter among different deletion plasmids and between control and leptin treatment (Figure 5B).

### 2.4. Site-Mutation Analysis of PPARα, PPARγ, and STAT3 on the Promoters of sf-1 and cyp17α

To further elucidate whether the regions of sf-1 and cyp17α promoters possessed PPARα, PPARγ, and STAT3 response elements, we performed the site-directed mutation at these regions of sf-1 and cyp17α promoters. Overexpressed PPARα reduced sf-1 promoter activity by 42% compared to the control, and its inhibitory effect was completely abolished when this site (−413/−396 bp) was mutated, suggesting that sf-1-PPARα (−413/−396 bp) site might inhibit PPARα-induced sf-1 transcription (Figure 6A). The sf-1 promoter activity was enhanced by 1.6-fold by overexpressed PPARγ, and mutation of sf-1-PPARγ (−1396/−1388 bp) site completely abolished the stimulatory effect of PPARγ, indicating that sf-1-PPARγ (−1396/−1388 bp) site enhanced PPARγ-induced sf-1 transcription (Figure 6B). STAT3 overexpression inhibited the promoter activity of sf-1 by 59% compared to the control. However, sf-1 promoter activity increased by 1.7-fold by mutation of sf-1-STAT3 (−1106/−1096 bp) site, suggesting that sf-1-STAT3 (−1106/−1096 bp) site down-regulated STAT3-induced sf-1 transcription (Figure 6C).

Overexpressed PPARα significantly reduced the cyp17α promoter activity by 48% compared to the control, whereas mutation of cyp17α-PPARα (−881/−864 bp) site significantly enhanced sf-1 promoter activity, suggesting that cyp17α-PPARα site (−881/−864 bp) down-regulated PPARα-induced cyp17α transcription (Figure 7A). Overexpressed PPARγ significantly increased sf-1 promoter activity, and mutation of cyp17α-PPARγ (−89/−70 bp) site completely abolished the stimulatory effect of PPARγ, indicating that cyp17α-PPARγ (−89/−70 bp) site played a positive regulatory role in cyp17α transcription (Figure 7B). Mutation of cyp17α-STAT3 (−1669/−1660 bp) significantly down-regulated STAT3-induced increase in cyp17α promoter activity, suggesting that cyp17α-STAT3 up-regulated STAT3-induced cyp17α transcription (Figure 7C).

To further determine whether the leptin-induced increase in sf-1 expression could be mediated by the STAT3 or PPARγ, we performed the site-directed mutation at the regions of sf-1 promoter and used leptin to incubate. Leptin incubation significantly increased sf-1 promoter activity. However, compared to the wild type pGl3−1981/+254 vector, the leptin-induced stimulatory effect was completely abolished after the mutation of sf-1-PPARγ (−1396/−1388 bp) site, sf-1−STAT3 (−1106/−1096 bp) site or sf-1-PPARγ&STAT3 (−1396/−1388 bp and −1106/−1096 bp) sites, respectively (Figure 8).

### 2.5. EMSA Analysis of Binding Sequence of Transcription Factors

We further used EMSA assays to explore whether PPARα, PPARγ, and STAT3 directly interact with sf-1 and cyp17α promoters. We made a series of probes by using biotin to label the sf-1-PPARα (−413/−396 bp), sf-1-PPARγ (−1396/−1388 bp), sf-1-STAT3 (−1106/−1096 bp), cyp17α-PPARα (−881/−864 bp), cyp17α-PPARγ (−89/−70 bp), and cyp17α-STAT3 (−1669/−1660 bp)-sequence, respectively. The results indicated that the sf-1-PPARα (−413/−396 bp) and sf-1-PPARγ (−1396/−1388 bp)-sequence did not compete for binding with the nuclear protein (Lane 3, Figure 9A,B), indicating that the sf-1-PPARα (−413/−396 bp) site and the sf-1-PPARγ (−1396/−1388 bp) site could not bind with the nuclear protein. For the sf-1-STAT3 site of sf-1 promoter, the unlabeled sf-1-STAT3 (−1106/−1096 bp)-sequence competed for the labeled probe (Lane 3, Figure 9C). Meanwhile, STAT3 treatment reduced the brightness of bands (Lane 5, Figure 9C), suggesting that the sf-1-STAT3 (−1106/−1096 bp) site could interact with sf-1 promoter and STAT3 weakened the binding process between STAT3 and the sf-1-STAT3 (−1106/−1096 bp) site of sf-1 promoter. Similarly, the unlabeled cyp17α-PPARα and cyp17α-STAT3 sites of cyp17α promoter did not compete for the labeled probe (Lane 3, Figure 9D,F), indicating that cyp17α-PPARα (−881/−864 bp) site and cyp17α-STAT3 (−1669/−1660 bp) site of cyp17α promoter were not bound with PPARα and STAT3, respectively. In addition, the unlabeled cyp17α-PPARγ (−89/−70 bp) site competed for the labeled probe (Lane 3, Figure 9E), while PPARγ treatment markedly increased the brightness of the binding band (Lane 5, Figure 9E), confirming that cyp17α-PPARγ (−89/−70 bp) was a binding site for PPARγ and PPARγ promoted the binding process between cyp17α-PPARγ and this binding site on cyp17α promoter. Taken together, our results demonstrated that sf-1 was the target gene of STAT3, and cyp17α was the target gene of PPARγ.

## 3. Discussion

In mammals, studies have indicated that the transcription of steroidogenic enzymes and steroid hormone receptors were mediated by transcription factor PPARs and STAT3 [12,26], but the direct molecular evidences were absent. The present study, for the first time, characterized the sequences of sf-1 and cyp17α promoters and elucidated their transcriptional regulatory mechanism in responses to PPARα, PPARγ, and STAT3.

In eukaryotes, the identification of the core promoter is the first step for exploring the mechanism of transcriptional initiation [27]. In the present study, the core promoters of sf-1 and cyp17α had different structures. For example, sf-1 had CCAAT box but no GC box, while cyp17α had GC box but no CCAAT box. Studies suggested that CCAAT box is the potential element that interacts with nuclear NF-Y on the sf-1 promoter [7,8]. The present study found that the proximal region of cyp17α promoter had TATA box but the proximal region of sf-1 promoter did not contain the TATA box, in agreement with other reports [2,7]. Usually, TATA-less promoters possessed various SP1 binding sites in their promoter regions [28] and similar result was found in the proximal promoter region of sf-1 in the present study. Wierstra [29] reported that SP1 directly bound with GC-rich domains and modulated transcription after various stimuli. Interestingly, we also found that the average activity of sf-1 promoter was higher than that of the cyp17α promoter. Thus, we speculated that SP1-rich and CCAAT-rich regions probable positively regulated promoter activity. In mammals, Scherrer et al. [8] reported that SP1 and CCAAT box positively regulated sf-1 promoter activity.

Identification of transcription factor binding sites (TFBS) plays an important role in deciphering the mechanisms of gene regulation [30]. PPARα and PPARγ are the two important nuclear transcription factors that regulate steroidogenesis [31,32]. In the present study, we found that PPARα significantly decreased the transcriptional activity of sf-1 and cyp17α promoter, while PPARγ up-regulated sf-1 and cyp17α transcription. These results indicated that PPARα and PPARγ differentially regulated steroidogenesis by targeting the sf-1 and cyp17α in yellow catfish. However, the EMSA analysis showed that PPARα and PPARγ sites from the sf-1 promoter were not functional, but PPARγ site from the cyp17α promoter was functional binding site, suggesting that PPARα and PPARγ indirectly mediated transcriptional activity of sf-1 and that PPARγ directly mediated transcriptional activity of cyp17α. STAT3 is considered the most important factor that transmits the leptin signal [21,22]. The activated STAT3 is capable of translocating to the nucleus to regulate gene transcription [33], and responsible for steroidogenic secretion [34]. In the present study, we found that overexpression of STAT3 negatively regulate sf-1 promoter activity. Similarly, studies suggested that STAT3 negatively regulated expression of sf-1 [35,36]. Furthermore, we also found that STAT3 positively regulated cyp17α promoter activity in yellow catfish. Roumaud and Martin [36] suggested that STAT3 negatively regulated expression of the cytochrome p450 side chain cleavage enzyme (p450scc) in leydig cells. However, the EMSA analysis showed that only STAT3 could bind with the sf-1 promoter but not with the cyp17α, suggesting that STAT3 directly mediated transcriptional activity of sf-1 and but indirectly mediated transcriptional activity of cyp17α.

Fish leptin is largely diverged from mammalian and it is well-known that the amino acid identity between yellow catfish leptin and human leptin is markedly low (almost 23%). However, as far as we know, fish leptin is hardly available commercially at present. Therefore, recombinant human leptin has widely been used to investigate the function of fish leptin [22,23,37,38]. Thus, in the present study, recombinant human leptin was used to study promoters function of sf-1 and cyp17α in yellow catfish. Leptin is known to have a significant impact on steroidogenesis in mammals. For example, Dhillon et al. [39] suggested that leptin directly activated sf-1 in mice. Similarly, our study indicated that 200 ng/mL leptin treatment up-regulated the transcription activity of sf-1 promoter indicated that leptin might regulate sf-1 promoter activity through STAT3 or PPARγ sites in yellow catfish. However, as another possibility, the obtained results in this study might be due to a pharmacological action of the “human” leptin [40,41]. Furthermore, we found that mutation of STAT3 binding site suppressed the leptin-induced increase in luciferase activity, suggesting that leptin might regulate sf-1 promoter activity through STAT3 site in yellow catfish. In addition, we also found that both of the mutation of STAT3 binding site and PPARγ binding site suppressed the leptin-induced increase in luciferase activity, suggesting that both STAT3 and PPARγ binding sites positively mediated sf-1 transcriptional response to leptin. Landry et al. [13] also suggested that the expression of steroidogenic genes was regulated by leptin through STAT in MA-10 leydig cells. Moreover, Reshma et al. [42] demonstrated a stimulatory effect of leptin on the mRNA expression of sf-1 in the ovarian tissues of Water Buffalo. However, we found that leptin had no effect on cyp17α promoter. To our best knowledge, currently no reports were available on the effects of leptin incubation on cyp17α expression and activity, and accordingly further investigation was needed to elucidate the mechanism.

In summary, the present study cloned and identified the different structure and function of sf-1 and cyp17α promoters. We found that three transcription factors (PPARα, PPARγ, and STAT3) regulated the transcription activities of sf-1 and cyp17α promoters and that the transcriptional activation of sf-1 was mediated by STAT3 and PPARγ under leptin signal.

## 4. Materials and Methods

Ethical statement: All animal experiments followed the ethical guidelines of Huazhong Agricultural University (HZAU) and were approved by the Animal Experimentation Ethics Committee of our university (Wuhan, Hubei, China) (identification code: Fish-2018-0827, Date: 27 August 2018).

### 4.1. Experimental Animals and Reagents

The yellow catfish individuals used for DNA and RNA extraction were bought from a local commercial farm (Wuhan, China). HEK293T cell lines were obtained from the Cell Resource Center of Fishery College of HZAU. Dulbecco’s Modified Eagles Medium (DMEM), fetal bovine serum (FBS) and 0.25% trypsin-EDTA were purchased from Gibco (USA). Recombinant human leptin (HPLC class), Penicillin, streptomycin and other reagents were purchased from Sigma (USA).

### 4.2. Promoter Cloning and Plasmids Construction

We identified the 5’ complementary DNA (cDNA) sequences and the (TSS) of sf-1 and cyp17α genes of yellow catfish by RNA ligase-mediated rapid amplification of 5’cDNA ends (RLM-5′RACE) method. The protocols of promoter cloning following those in Xu et al. [43]. For generating the luciferase reporter constructs, we subcloned different plasmids with sf-1 and cyp17α promoters into pGl3-Basic vectors (Promega, USA) by using Sac I and Hind III restriction sites. The plasmids of PPARα, PPARγ, and STAT3 overexpression were subcloned into pcDNA3.1 (+) vector with FLAG-tag sequence inserted into the C-terminus of PPARα, PPARγ, and STAT3 sequences. We used ClonExpress II One Step Cloning Kit (Vazyme, Piscataway, NJ, USA) to ligate all of the products. Primers used in 5’RACE and plasmid construction are presented in Appendix A.

On the basic of the distance from its TSS, we named the plasmids as pGl3−1981/+254 sf-1 promoter and pGl3−2034/+68 cyp17α promoter, respectively. Using the templates of pGl3−1981/+254 of sf-1 vector, we produced the plasmids of pGl3−489/+254, pGl3−921/+254, pGl3−1421/+254, and pGl3−1981/+254 of sf-1. Similarly, the plasmids of pGl3−330/+68, pGl3−687/+68, pGl3−1402/+68, and pGl3−2034/+68 of cyp17α were generated through using pGl3−2034/+68 of cyp17α vector as the template.

### 4.3. Sequence Analysis

We compared nucleotide sequences of sf-1 and cyp17α promoters with DNA sequences from the GenBank database. In order to analyze sf-1 and cyp17α promoters, we predicted putative TFBS by MatInspector online (http://www.genomatix.de/) and the JASPAR database (http://jaspar.genereg.net/). We conducted sequence alignments with the Clustal-W multiple alignment algorithm.

### 4.4. Plasmid Transfections and Assays of Luciferase Activities

We used Lipofectamine 2000 (Invitrogen, Carlsbad, CA, USA) to transiently transfect the plasmids into HEK293T cells following the manufacture’s protocols. We used all reporter plasmids in equimolar amounts in Opti-MEM (Invitrogen, USA), and 20 ng pRL-TK were co-transfected as the control. At 4 h after the transfection, we replaced the transfection medium by DMEM (10% FBS) with or without 200 ng/mL leptin. The leptin concentrations were selected after the publications in our laboratory [22,37]. After 24 h incubation, we harvested cells to assay the luciferase activity by Dual-Luciferase Reporter Assay System (Promega, USA). The relative luciferase activities of these promoters were obtained by the ratio of Firefly to Renilla luciferase activity. We conducted all experiments in triplicates.

### 4.5. Protein Expression of Yellow Catfish PPARα, PPARγ, and STAT3 in HEK293T Cells

We used western blot to determine the protein expression of yellow catfish PPARα, PPARγ, and STAT3 in HEK293T cells based on the methods described in our studies [44,45]. The primary antibodies included rabbit polyclonal of anti-GAPDH (1:10.000, Abcam, Cambridge, UK) and anti-FLAG (1:10.000, Proteintech, Wuhan, china). We visualized the protein bands by Vilber Fusion FX6 Spectra imaging system (Vilber Lourmat, Collegien, France), and quantified them by Image-Pro Plus 6.0.

### 4.6. Site-Mutation Analysis of Binding Sites on the sf-1 and cyp17α Promoters

To identify the corresponding binding sites on the regions of sf-1 and cyp17α promoters, we used QuickChange II Site-Directed Mutagenesis Kit (Vazyme, Piscataway, NJ, USA) to perform site-directed mutagenesis analysis. pGl3-sf-1-1981 and pGl3-cyp17α-2034 were used as templates, respectively. The primers for mutagenesis are shown in Appendix A. We name these mutant constructs as Mut-sf-1-PPARα, Mut-sf-1-PPARγ, Mut-sf-1-STAT3, Mut-sf-1-PPARγ&STAT3, Mut-cyp17α-PPARα, Mut-cyp17α-PPARγ, and Mut-cyp17α-STAT3, respectively. Then we used Lipofectamine 2000 (Invitrogen) to co-transfect the plasmids and pRL-TK into HEK293T cells. After 4 h, we replaced the transfection medium by DMEM (10% FBS) with or without 200 ng/mL leptin. After 24 h incubation, we harvested cells to perform the analysis of the luciferase activity, based on the procedure described above.

### 4.7. Electrophoretic Mobility-Shift Assay (EMSA)

We conducted EMSA assays to explore the functional binding sites of PPARα, PPARγ, and STAT3 on the regions of sf-1 and cyp17α promoters, based on the methods of Xu et al. [43]. We transfected HEK293T cells with the same amount of pcDNA3.1, PPARα, PPARγ, and STAT3 plasmid, respectively. We extracted the nuclear proteins from HEK293T cells and measured the concentrations of proteins by the BCA method. All the oligonucleotide sequences for EMSA analysis are presented in Appendix A.

### 4.8. Statistical Analysis

We used SPSS 19.0 software for all these statistical analysis. All of these data were expressed as means ± SEM (standard errors of means). Before statistical analysis, we evaluated all data for normality using the Shapiro–Wilk test. In order to test the homogeneity of variances, we performed Bartlett’s test. We analyzed data with one-way ANOVA and Duncan’s multiple or Student’s t-test where appropriate. Significant level was *p* < 0.05.

## Figures and Tables

**Figure 1 ijms-22-00195-f001:**
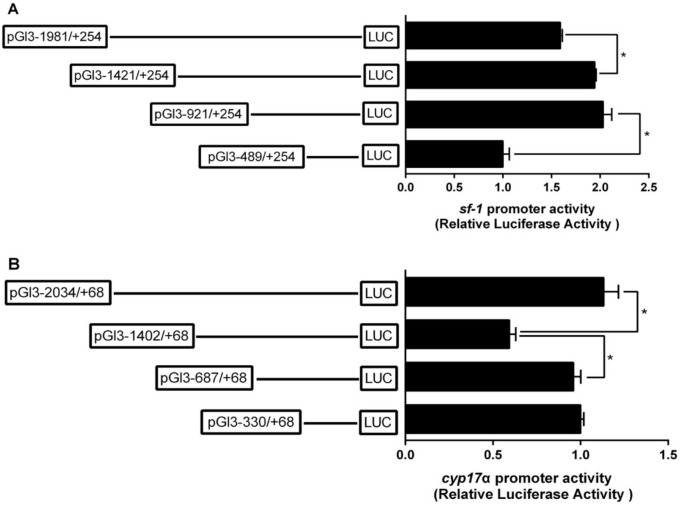
5’ unidirectional deletion assays of the sf-1 and cyp17α promoters of yellow catfish. Schematic diagram of truncated promoters are shown at the left panel. The results for corresponding luciferase reporter assay are shown at right panel. (**A**) A series of plasmids containing 5’ unidirectional deletions of the sf-1 promoter regions (pGl3-1981, −1421, −921, and −489) fused in frame to the luciferase gene were transfected into HEK293T cells; (**B**) A series of plasmids containing 5’ unidirectional deletions of the cyp17α promoter regions (pGl3-2034, −1402, −687, and −330) fused in frame to the luciferase gene were transfected into HEK293T cells. Values represent the ratio between firefly and Renilla luciferase activities. Results are expressed as the mean ± SEM arbitrary units of three independent experiments. Symbol (*) indicates significant differences between the two groups (*p* < 0.05).

**Figure 2 ijms-22-00195-f002:**
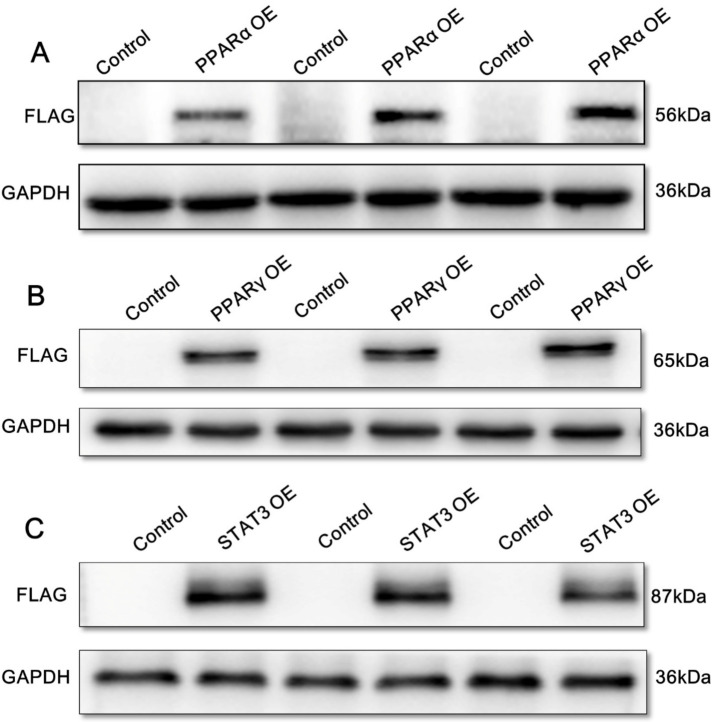
Overexpression (OE) of yellow catfish PPARα, PPARγ, and STAT3 in HEK293T cells after Western blot analysis using FLAG antibody. (**A**) PPARα over-expression (OE); (**B**) PPARγ OE; (**C**) STAT3 OE. GAPDH represents glyceraldehyde-3-phosphate dehydrogenase. (*n* = 3 represents three independent biological experiments).

**Figure 3 ijms-22-00195-f003:**
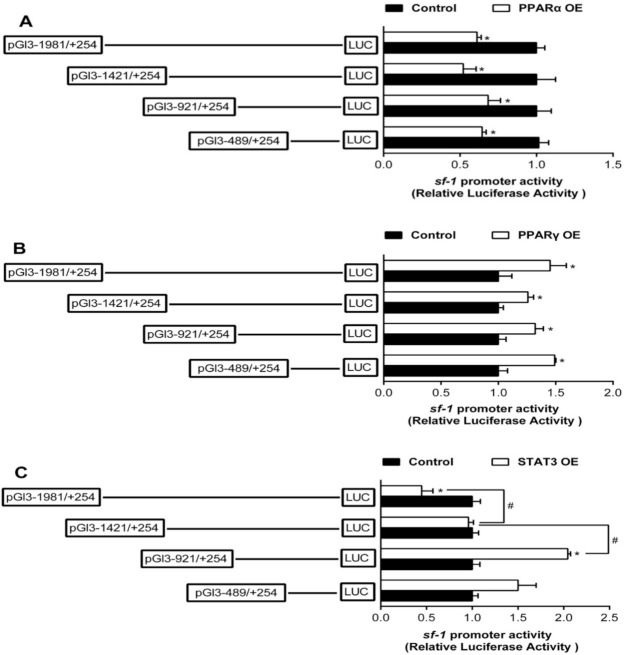
Overexpression (OE) analysis of 5’ unidirectional deletion assays of the sf-1 promoters of yellow catfish. (**A**) PPARα OE; (**B**) PPARγ OE; (**C**) STAT3 OE. Values are presented as mean ± SEM (*n* = 3 independent biological experiments). Asterisk (*) indicates significant differences in relative luciferase activities between the PPARα-, PPARγ-, and STAT3-OE group and the control (*p* < 0.05). Hash symbol (#) indicates significant difference between the same OE groups with different deletion regions (*p* < 0.05). The relative luciferase activity was presented as the fold activated by PPARα, PPARγ, and STAT3 compared with the control, respectively.

**Figure 4 ijms-22-00195-f004:**
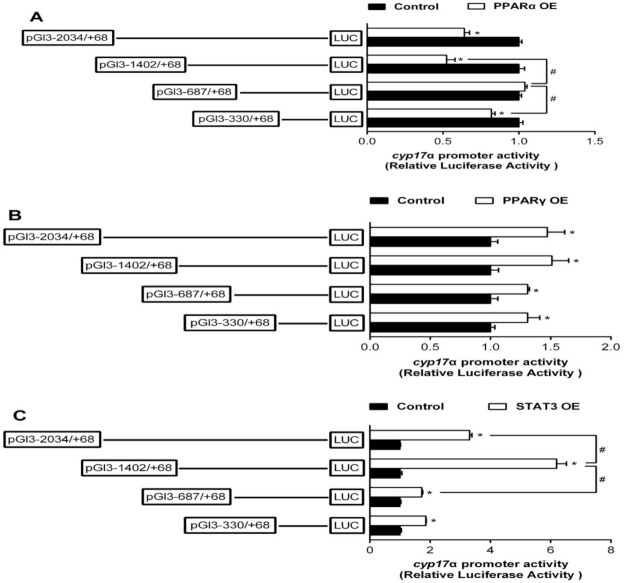
Overexpression (OE) analysis of 5′ unidirectional deletion of the cyp17α promoters of yellow catfish. (**A**) PPARα OE; (**B**) PPARγ OE; (**C**) STAT3 OE. Values are presented as mean ± SEM (*n* = 3 independent biological experiments). Asterisk (*) indicates significant differences in relative luciferase activities between the PPARα-, PPARγ-, and STAT3-OE group and the control (*p* < 0.05). Hash symbol (#) indicates significant difference between the same OE groups with different deletion regions (*p* < 0.05). The relative luciferase activity was presented as the fold activated by PPARα, PPARγ, and STAT3 compared with the control, respectively.

**Figure 5 ijms-22-00195-f005:**
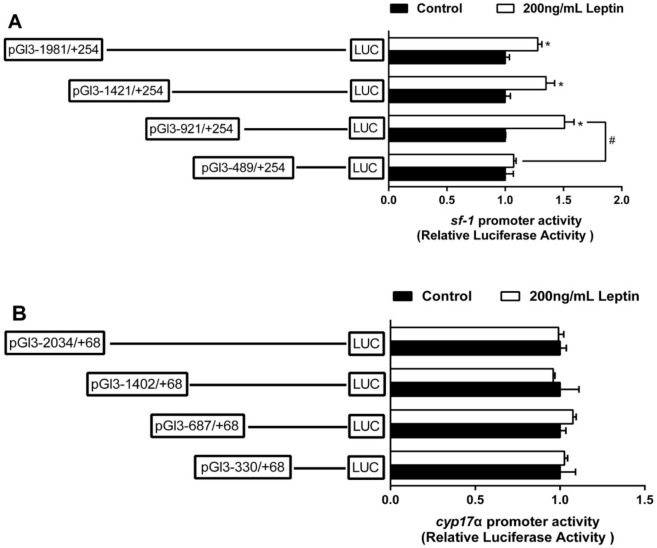
5’ unidirectional deletion assays for sf-1 (**A**) and cyp17α (**B**) promoters after 200 ng/mL leptin treatment for 24 h. Values showed the ratio of activities of firefly to Renilla luciferase, normalized to the control. Results were presented as mean ± SEM (*n* = 3 independent biological experiments). Asterisk (*) indicates significant differences between two treatments in the same plasmid (*p* < 0.05). Hash symbol (#) indicates significant differences between two 5′ unidirectional deletion plasmids under the same treatment (*p* < 0.05).

**Figure 6 ijms-22-00195-f006:**
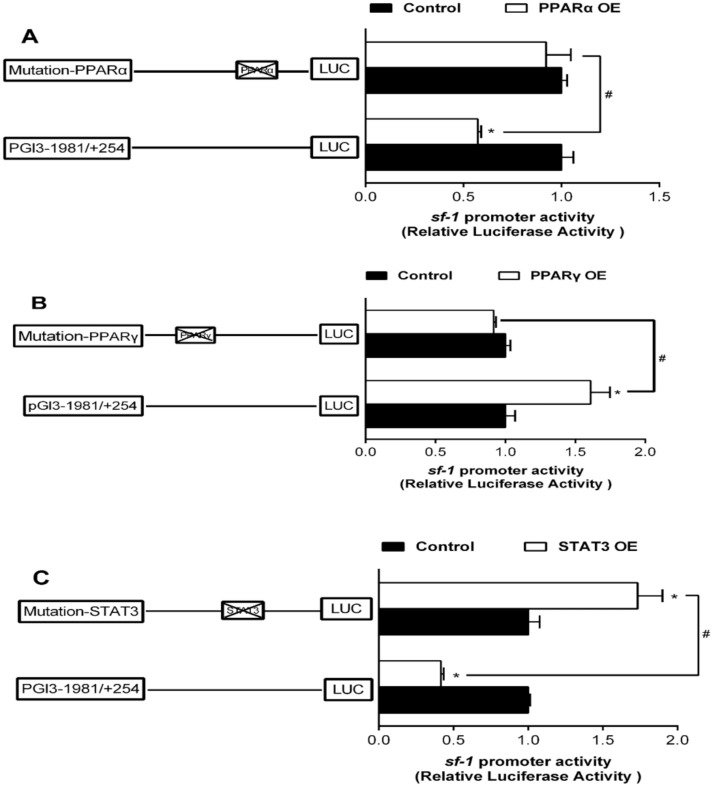
Assays of sf-1 promoter activities by mutagenesis on predicted PPARα, PPARγ, and STAT3 binding sites at 24 h. (**A**) Site mutagenesis of PPARα on −1981/+254 sf-1 promoter; (**B**) Site mutagenesis of PPARγ on −1981/+254 sf-1 promoter; (**C**) Site mutagenesis of STAT3 on −1981/+254 sf-1 promoter. Values are presented as mean ±SEM (*n* = 3 independent biological experiments). Asterisk (*) indicates significant differences between PPARα, PPARγ, and STAT3 overexpression and the control (*p* < 0.05). Hash symbol (#) indicates significant differences between two 5′ unidirectional deletion plasmids under the same treatment (*p* < 0.05).

**Figure 7 ijms-22-00195-f007:**
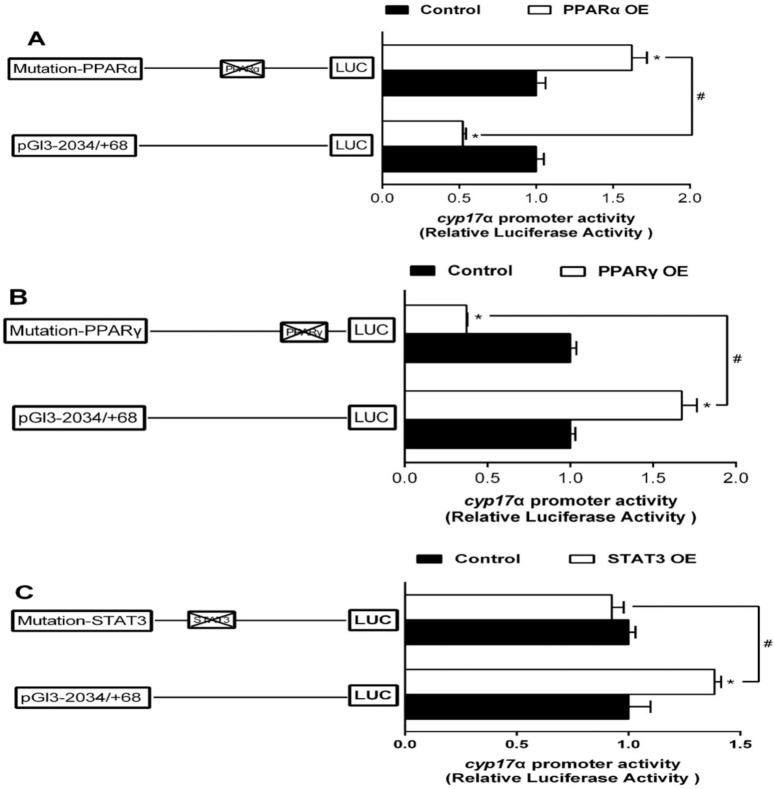
Assays of cyp17α promoter activities of mutagenesis on predicted PPARα, PPARγ, and STAT3 binding sites at 24 h. (**A**) Site mutagenesis of PPARα on −2034/+68 cyp17α promoter; (**B**) Site mutagenesis of PPARγ on −2034/+68 cyp17α promoter; (**C**) Site mutagenesis of STAT3 on −2034/+68 cyp17α promoter. Values are presented as mean ± SEM (*n* = 3 independent biological experiments). Asterisk (*) indicates significant differences between PPARα, PPARγ, and STAT3 overexpression and the control (*p* < 0.05). Hash symbol (#) indicates significant differences between two 5′ unidirectional deletion plasmids under the same treatment (*p* < 0.05).

**Figure 8 ijms-22-00195-f008:**
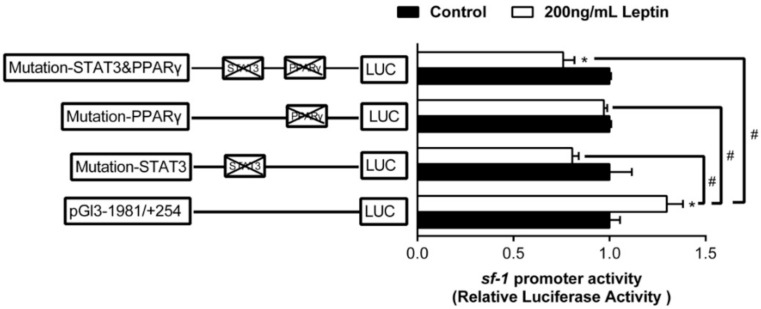
Assays of sf-1 promoter activity of site mutagenesis on predicted PPARγ and STAT3 after leptin treatment at 24 h. Values are presented as mean ± SEM (*n* = 3 independent biological experiments). Asterisk (*) indicates significant differences between different treatments in the same plasmid (*p* < 0.05). Hash symbol (#) indicates significant differences between different 5’ unidirectional deletion plasmids under the same treatment (*p* < 0.05).

**Figure 9 ijms-22-00195-f009:**
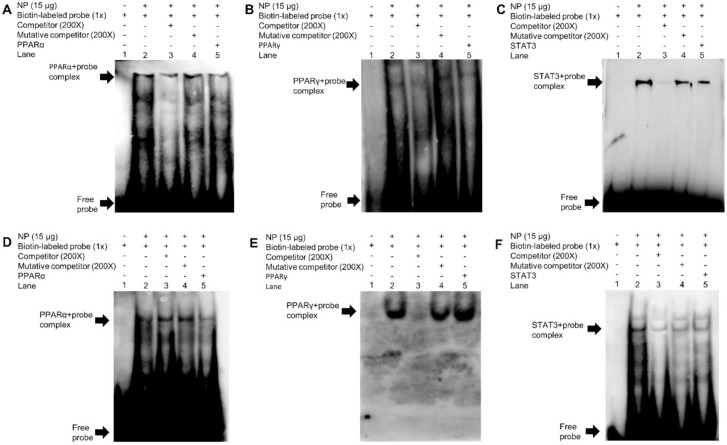
EMSA analysis of predicted PPARα, PPARγ, and STAT3 binding sf-1 and cyp17α promoters. (**A**) −413/−396 bp PPARα binding site of sf-1 (sf-1-PPARα); (**B**) −1396/−1388 bp PPARγ binding site of sf-1 (sf-1-PPARγ); (**C**) −1106/−1096 bp STAT3 binding site of sf-1 (sf-1-STAT3); (**D**) −881/−864 bp PPARα binding site of cyp17α (cyp17α-PPARα); (**E**) −89/−70 bp PPARγ binding site of cyp17α (cyp17α-PPARγ); (**F**) −1669/−1660 bp STAT3 binding site of cyp17α (cyp17α-STAT3). NP represents nuclear protein.

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
