# Peer review of "Functional Analysis of Steroidogenic Factor 1 (sf-1) and 17α-Hydroxylase/Lyase (cyp17α) Promoters in Yellow Catfish Pelteobagrus fulvidraco"

_ijms, 2020, doi:10.3390/ijms22010195_

Round 1

Reviewer 1 Report

The proposed manuscript (ms) contributes to functional understanding of steroidogenesis and its regulation. Interestingly, authors selected genes sf-1 and cyp17α that cooperate with other transcriptional factors. I have not found any fundamental issues in proposed ms. The ms is technically sound, presented in an intelligible fashion and written in good-quality English. In my opinion, some parts can be improved for better understanding and making ms more interesting to read. Some typos were found.

I have some minor suggestions such as:

1) Luciferase is an important component of the ms but it is not introduced in the introduction section. How sf-1/cyp17α promoter activity corresponds with luciferase activity? Why did you use luciferase in your study?

2) Line 13-17: In the sentence "We successfully obtained ...", "respectively" is missing.

3) Including a vital role in reproduction (the line 47) and signaling in mammals, leptin has also important function in digestion (e.g. in birds). The connection with digestion was revealed by expression and immunohistochemistry analysis by Seroussi et al (2019). The study showed that leptin is expressed in chicken intestine (duodenum). Could authors involve this interesting study into the introduction section (or discussion)? Link for Seroussi et al. paper is: https://www.mdpi.com/1422-0067/20/18/4489/htm.

4) The line 66: The TSS abbreviation is not explained after first use. It is explained further in the M&M section. 

5) Figure 2A, upper line label is "FIAG". Is it correct?

6) Figure 2, the label/abbreviation GAPDH is not explained.

7) Figure 2, there is each gene and control for three times on the figure? Why? Maybe it could be indicated in figure description.

8) The line 145: There is "of of" in a sentence.

9) Some sentences in the section 2.5 are difficult to read for me. For example, "The results indicated that the 200-fold unlabeled sf-1-PPARα and sf-1-PPARγ-sequence did not compete the labeled probe, indicating that sf-1-PPARα (-413/-396 bp) site and sf-1-PPARγ (-1396/-1388 bp) site could not be bound by PPARα and PPARγ, respectively." Maybe if you try to explain a description of the figure 9 more in detail, the section 2.5. would be easy to read.

10) Figure 9: NP abbreviation is not explained.

11) The line 262: The TFBS abbreviation is not explained after first use. It is explained further in the M&M section. 

12) The line 276: Check if a dot in citration Roumaud and Martin. [33] (behind Martin) is used correctly.

13) Lines 302-3: I would suggest: "The yellow catfish individuals used for DNA and RNA extraction were obtained/brought/bought (or originate) from a local commercial farm (Wuhan, China)." As well as in the sentence "HEK293T cell lines were obtained/brought/bought (or established) from ..."

14) Lines 349-352: Double-check if "respectively" terms are necessary to use in sentences, and check the format of labels of mutant construct (Mut-sf-1-PPARγ & STAT3 vs. Mut-cyp17α-STAT3).

15) The line 428: There is a typo "23. 23."

Literature:

Seroussi, E.; Knytl, M.; Pitel, F.; Elleder, D.; Krylov, V.; Leroux, S.; Morisson, M.; Yosefi, S.; Miyara, S.; Ganesan, S.; Ruzal, M.; Andersson, L.; Friedman-Einat, M. Avian Expression Patterns and Genomic Mapping Implicate Leptin in Digestion and TNF in Immunity, Suggesting That Their Interacting Adipokine Role Has Been Acquired Only in Mammals. Int. J. Mol. Sci. 201920, 4489.

Author Response

Reviewer 1:

Comment 1: The proposed manuscript (ms) contributes to functional understanding of steroidogenesis and its regulation. Interestingly, authors selected genes sf-1 and cyp17α that cooperate with other transcriptional factors. I have not found any fundamental issues in proposed ms. The ms is technically sound, presented in an intelligible fashion and written in good-quality English. In my opinion, some parts can be improved for better understanding and making ms more interesting to read. Some typos were found.

Response 1: Thank you very much for your constructive comments. We have strictly revised our manuscript based on your important comments and listed our responses on a point-to-point basis on your comments.

Comment 2: I have some minor suggestions such as: 1) Luciferase is an important component of the ms but it is not introduced in the introduction section. How sf-1/cyp17α promoter activity corresponds with luciferase activity? Why did you use luciferase in your study?

Response 2: Thank you very much for your comments to our manuscript. The relevant description of luciferase was added in the introduction section. We subcloned different plasmids with sf-1 and cyp17α promoters into pGl3-Basic vectors, so the promoter activity can be detected by luciferase activity. It is well-known that the application of luciferase assay is a well-established technique in molecular biology for analysis of cloned promoter DNA fragments (Solberg and Krauss, 2013; Delhove et al., 2017). Thus, luciferase assay was used in our study.

References:

Delhove, J.M.K.M.; Karda, R.; Hawkins, K.E.; FitzPatrick, L.M.; Waddington, S.N.; McKay, T.R. Bioluminescence Monitoring of Promoter Activity In Vitro and In Vivo. Methods. Mol. Biol. 2017, 1651, 49-64.

Solberg, N.; Krauss, S. Luciferase assay to study the activity of a cloned promoter DNA fragment. Methods. Mol. Biol. 2013, 977, 65-78.

Comment 3: 2) Line 13-17: In the sentence "We successfully obtained ...", "respectively" is missing.

Response 3: We have added "respectively" in line 13-17 of our manuscript based on your constructive suggestion.

Comment 4: 3) Including a vital role in reproduction (the line 47) and signaling in mammals, leptin has also important function in digestion (e.g. in birds). The connection with digestion was revealed by expression and immunohistochemistry analysis by Seroussi et al (2019). The study showed that leptin is expressed in chicken intestine (duodenum). Could authors involve this interesting study into the introduction section (or discussion)? Link for Seroussi et al. paper is: https://www.mdpi.com/1422-0067/20/18/4489/htm.

Response 4: We have cited the interesting study by Seroussi et al (2019) into the introduction section of our manuscript based on your constructive suggestion.

Comment 5: 4) The line 66: The TSS abbreviation is not explained after first use. It is explained further in the M&M section. 

Response 5: We have changed "TSS" to “transcription start sites (TSS)” in the line 70 in our manuscript based on your constructive suggestion.

Comment 6: 5) Figure 2A, upper line label is "FIAG". Is it correct?

Response 6: Sorry and a mistake was made here. We have changed the upper line label of Figure 2A from "FIAG " to "FLAG".

Comment 7: 6) Figure 2, the label/abbreviation GAPDH is not explained.

Response 7: The label/abbreviation GAPDH has been explained in Figure 2 based on your constructive suggestion.

Comment 8: 7) Figure 2, there is each gene and control for three times on the figure? Why? Maybe it could be indicated in figure description.

Response 8: In Figure 2, n=3 represents three independent biological experiments. It has been indicated in figure 2 description.

Comment 9: 8) The line 145: There is "of of" in a sentence.

Response 9: We have removed one of "of" in a sentence of line 150 based on your constructive suggestion.

Comment 10: 9) Some sentences in the section 2.5 are difficult to read for me. For example, "The results indicated that the 200-fold unlabeled sf-1-PPARα and sf-1-PPARγ-sequence did not compete the labeled probe, indicating that sf-1-PPARα (-413/-396 bp) site and sf-1-PPARγ (-1396/-1388 bp) site could not be bound by PPARα and PPARγ, respectively." Maybe if you try to explain a description of the figure 9 more in detail, the section 2.5. would be easy to read.

Response 10: We revised these sentences you mentioned. For details, please see the text.

Comment 11: 10) Figure 9: NP abbreviation is not explained.

Response 11: NP abbreviation has been explained in Figure 9.

Comment 12: 11) The line 262: The TFBS abbreviation is not explained after first use. It is explained further in the M&M section. 

Response 12: We have revised our manuscript based on your constructive suggestion. For details, please see the line 271 and 348 of the text.

Comment 13: 12) The line 276: Check if a dot in citration Roumaud and Martin. [33] (behind Martin) is used correctly.

Response 13: We have removed a dot in citration Roumaud and Martin. [33] (behind Martin) in the line 286.

Comment 14: 13) Lines 302-3: I would suggest: "The yellow catfish individuals used for DNA and RNA extraction were obtained/brought/bought (or originate) from a local commercial farm (Wuhan, China)." As well as in the sentence "HEK293T cell lines were obtained/brought/bought (or established) from ..."

Response 14: We have revised our manuscript based on your constructive suggestion. For details, please see lines 324-325 of the text.

Comment 15: 14) Lines 349-352: Double-check if "respectively" terms are necessary to use in sentences, and check the format of labels of mutant construct (Mut-sf-1-PPARγ & STAT3 vs. Mut-cyp17α-STAT3).

Response 15: We have carefully checked the sentences and the format of labels you mentioned and revised our manuscript based on your constructive suggestion. For details, please see the text.

Comment 16: 15) The line 428: There is a typo "23. 23."

Response 16: Sorry and mistake was made here. We have revised our manuscript based on your comments.

Reviewer 2 Report

This manuscript characterizes the structures and functions of sf-1 and cyp17a in yellow catfish. The paper is well written, most of the experimental protocols are sound, and the results well presented. I believe this is a valuable study offering novel insights into the transcriptional regulation of sf-1 and cyp17a in yellow catfish. Please consider following suggestion to improve the authors contribution on the research field.

L305: “Recombinant human leptin”

It is well known that fish leptin is largely diverged from mammalian leptin; the amino acid identity between fish leptin and human leptin is markedly low (almost 15 %). This point needs to be mentioned in the manuscript and add some discussion. As mention in the manuscript, leptin might regulate sf-1 promoter activity through STAT3 site in yellowtail. However, as another possibility, the obtained results in this study might be due to a pharmacological action of the “human” leptin.

Author Response

Reviewer 2:

Comment 1: This manuscript characterizes the structures and functions of sf-1 and cyp17α in yellow catfish. The paper is well written, most of the experimental protocols are sound, and the results well presented. I believe this is a valuable study offering novel insights into the transcriptional regulation of sf-1 and cyp17α in yellow catfish. Please consider following suggestion to improve the authors contribution on the research field.

Response 1: Thank you very much for your help and valuable comments to our manuscript. We have revised our manuscript and listed our responses based on your important comments.

Comment 2: L305: “Recombinant human leptin”: It is well known that fish leptin is largely diverged from mammalian leptin; the amino acid identity between fish leptin and human leptin is markedly low (almost 15 %). This point needs to be mentioned in the manuscript and add some discussion. As mention in the manuscript, leptin might regulate sf-1 promoter activity through STAT3 site in yellowtail. However, as another possibility, the obtained results in this study might be due to a pharmacological action of the “human” leptin.

Response 2: Thank you very much for your constructive comments. We have strictly revised our manuscript based on your important comments. Indeed, the amino acid identity between yellow catfish leptin and human leptin is markedly low (almost 23 %). However, as far as we know, commercial fish leptin, including yellow catfish leptin, is hardly available at present. On the other hand, many studies used mammalian recombinant leptin (including vivo injection or vitro cell incubation of fish) to assess leptin function of fish (Peyon et al., 2001; Weil et al., 2003; de Pedro et al., 2006; Gorissen et al., 2012; Zhang et al., 2017). Thus, in the preset study, recombinant human leptin was used to study promoter function of sf-1 and cyp17a in yellow catfish. We have added relevant content to the discussion in the original article to explain the results of this part. Please see the revised document for details.

As you mentioned in the manuscript, our study showed that leptin treatment up-regulated the transcription activity of sf-1 promoter, indicating that leptin might regulate sf-1 promoter activity through STAT3 site in yellow catfish. However, as another possibility, the obtained results in this study might be due to a pharmacological action of the “human” leptin. Furthermore, we found that mutation of STAT3 binding site suppressed the leptin-induced increase of luciferase activity, suggesting that leptin might regulate sf-1 promoter activity through STAT3 site in yellow catfish. We have added these statements to the Discussion section. For details, pls. see the text.

References:

de Pedro, N.; Martínez-Alvarez, R.; Delgado, M.J. Acute and chronic leptin reduces food intake and body weight in goldfish (Carassius auratus). J. Endocrinol. 2006, 188, 513-520.

Gorissen, M.; Bernier, N.J.; Manuel, R.; de Gelder, S.; Metz, J.R.; Huising, M.O.; Flik, G. Recombinant human leptin attenuates stress axis activity in common carp (Cyprinus carpio L.). Gen. Comp. Endocrinol. 2012, 178, 75-81.

Peyon, P.; Zanuy, S.; Carrillo, M. Action of leptin on in vitro luteinizing hormone release in the European sea bass (Dicentrarchus labrax). Biol. Reprod. 2001, 65, 1573-1578.

Weil, C.; Le Bail, P.Y.; Sabin, N.; Le Gac, F. In vitro action of leptin on FSH and LH production in rainbow trout (Onchorynchus mykiss) at different stages of the sexual cycle. Gen. Comp. Endocrinol. 2003, 130, 2-12.

Zhang, Y.; Chua Jr, S. Leptin Function and Regulation. Compr. Physiol. 2017, 8, 351-369.

Round 2

Reviewer 2 Report

Well revised. No further comment form me.